# OpenReview forum: "Momentum Steering: Activation Steering Meets Optimization"
_ICLR.cc/2026/Conference — Submitted to ICLR 2026_

### Official Review · Reviewer_VGaF · 2025-10-31

**Soundness:** 3
**Presentation:** 2
**Contribution:** 3
**Rating:** 6
**Confidence:** 4

**Summary:**

The authors reframe activation steering in LLMs as an optimization problem and propose momentum-based steering that incorporates momentum from gradient-based optimization into the computation of steering vectors. Rather than using simple difference-in-means between contrastive prompts, they accumulate these differences across layers via momentum updates. The authors show that popular methods like ActAdd and Mean-AcT can be derived as (projected) gradient descent updates in their framework, then extend this framework with momentum and an Adam-based variants. Experiments on jailbreaking and toxicity mitigation tasks show improvements over baselines across multiple model families.

**Strengths:**

- reinterpreting activation steering as PGD is creative and provides a principled framework for understanding existing methods
- consistent improvements across diverse models (Qwen, Gemma, Llama with 3b-32b params) and tasks
- seems easy to integrate with existing steering methods without significant computational overhead
- provides stability analysis and connects to well-established optimization literature

**Weaknesses:**

1. Poor presentation: multiple typos (e.g., "zero k" line 233), inconsistent notation, and unclear writing throughout
  - switches between $x(k)$ and $x(t)$ without explanation; unclear distinction between $\mathbf{q}$ and $\mathbf{p}$ in sections 2.1-2.2; unexplained notation like $k=[K]$
  - see questions for more unclarity
2. no statistical significance shown: no error bars on experiments, making it impossible to assess significance given that results are expected to have significant variance
3. Questionable experimental choices: randomly initialized model uses unrealistic architecture (150 layers), why not use actual trained models for validation?

**Questions:**

1. Am I correct in understanding that the calculations for the steering functions (including momentum-based updates) are only when constructing the steering vectors -- when applying the steering vectors, the calculations stay the same as in ActAdd? Or are the you also changing the calculations done during inference?
2. Why choose Bregman divergence over other divergence functions?
3. Can you provide intuition about how different choices of $h$ in the Bregman divergence lead to different feature maps and steering behaviors?
4. How does this relate to matrix-based steering methods ([Postmus & Abreu, 2025](https://arxiv.org/abs/2410.16314)) or flow-based methods ([Rodríguez et al, 2024](https://arxiv.org/html/2410.23054v1))?
5. For the randomly initialized model experiments (Appendix B):
   - Why use such unrealistic architecture (150 layers)?
   - Why not validate on actual trained models?
   - Would the trends persist from initialization to final trained models?
6. In equation 2.3, are you averaging over time? At which tokens do you extract activations and where do you apply them?
7. What explains the inconsistent notation between $x(k)$ and $x(t)$, and between $\mathbf{q}$ and $\mathbf{p}$?
8. What does $k=[K]$ mean before equations 1 and 2?

---

> ### Author Response · Authors · 2025-11-19
> **Response to Reviewer VGaF (1)**
>
> Thank you for your thoughtful review and valuable feedback. Below, we address your concerns.
>
> -----
>
> **Q1. Poor presentation: multiple typos (e.g., "zero k" line 233), inconsistent notation, and unclear writing throughout.**
> **- Switches between $x(k)$ and $x(t)$ without explanation; unclear distinction between $q$ and $p$ in sections 2.1-2.2; What explains the inconsistent notation between $x(k)$ and $x(t)$, and between $q$ and $p$?**
> **- Unexplained notation like $k=[K]$. What does $k=[K]$ mean before equations 1 and 2?**
>
> **Answer:** Thank you for your suggestions. We have revised Sections 2.2, 2.3, and 3.2 to improve the paper's presentation and clarify our optimization framework for activation steering. We also respond to your specific concerns below.
>
> **Switches between $x(k)$ and $x(t)$ without explanation; unclear distinction between $q$ and $p$ in sections 2.1-2.2; What explains the inconsistent notation between $x(k)$ and $x(t)$, and between $q$ and $p$?**
>
> We use $x(t)$ to denote continuous-time activations and $x(k)$ for their discrete-time counterparts. The discretization from $x(t)$ to $x(k)$ is explained in Lines 171--172 of the original manuscript and Lines 184-185 of the revised version.
>
> Regarding the distinction between $q$ and $p$ in Sections 2.1--2.2, these were indeed typos. We are grateful to the reviewer for pointing them out, and in the revised manuscript, we have replaced all occurrences of $q$ with $p$ to denote the input prompt consistently.
>
> **Unexplained notation like $k=[K]$. What does $k=[K]$ mean before equations 1 and 2?**
>
> $k=[K]$ means for $k=1, 2, \dots, K$. We adopt this notation from previous works such as [1,2].
>
> **References**
>
> [1] Andy Arditi, Oscar Balcells Obeso, Aaquib Syed, Daniel Paleka, Nina Rimsky, Wes Gurnee, and Neel Nanda. Refusal in language models is mediated by a single direction. NeurIPS, 2024.
>
> [2] Bruce W Lee, Inkit Padhi, Karthikeyan Natesan Ramamurthy, Erik Miehling, Pierre Dognin, Manish Nagireddy, and Amit Dhurandhar. Programming refusal with conditional activation steering. ICLR, 2025.
>
> **Q2. No statistical significance shown: no error bars on experiments, making it impossible to assess significance given that results are expected to have significant variance**
>
> **Answer:** Thank you for pointing this out. However, we would like to note that for our jailbreaking experiments, we follow the setup proposed in [1], in which the train and test sets are fixed. For that experiment, our model generations are performed using greedy decoding, consistent with the setup in [1], making our experiments deterministic. For the toxicity mitigation experiment, we closely follow the setup in [2], where model generations are performed with greedy decoding. However, in this setting, the test prompts are randomly selected instead [2]. To account for this, the experiments were conducted over 10 runs, and the mean and standard deviations are reported in Table 4 in both the original and revised manuscripts.
>
> In addition, we are attempting to run an experiment on the jailbreaking task using TopK sampling instead of greedy decoding, and we will provide this result in our next reply.
>
> **References**
>
> [1] Hieu M. Vu and Tan Minh Nguyen. Angular steering: Behavior control via rotation in activation space. NeurIPS, 2025.
>
> [2] Pau Rodriguez, Arno Blaas, Michal Klein, Luca Zappella, Nicholas Apostoloff, marco cuturi, and Xavier Suau. Controlling language and diffusion models by transporting activations. ICLR, 2025.

---

> ### Author Response · Authors · 2025-11-19
> **Response to Reviewer VGaF (2)**
>
> **Q3. Questionable experimental choices: randomly initialized model uses unrealistic architecture (150 layers), why not use actual trained models for validation?**
>
> **For the randomly initialized model experiments (Appendix B):
> Why use such unrealistic architecture (150 layers)?
> Why not validate on actual trained models?
> Would the trends persist from initialization to final trained models?**
>
> **Answer:** The convergence of gradient descent and momentum-based accelerated gradient descent only becomes apparent after a sufficient number of update steps. In our optimization framework for activation steering in LLMs, the number of update steps corresponds to the number of layers in the model. However, existing pretrained LLM architectures do not have enough layers to clearly exhibit convergence of the steering process. For example, Qwen2.5-14B-Instruct, Llama3.2-3B-Instruct, and Gemma2-9B-Instruct contain only 48, 28, and 42 layers, respectively.
>
> To empirically demonstrate the convergence of the steering process and support the correspondence between activation steering and PGD, we therefore use a randomly initialized LLM in which we can increase the number of layers (150 layers). This provides enough gradient descent steps for the gradient norm (i.e., the candidate direction norm in this setting; see the explanation in Section 2 of our manuscript) to approach zero. Figure 2 in our manuscript illustrates this convergence.
>
> For comparison, we also report an analogous result for the pretrained Llama3.2-3B-Instruct and Llama3.1-38-Instruct models in Figures 4 and 5 (Appendix G) of our revision. Overall, we still observe that the gradient norm (candidate direction norm) decreases as the number of layers increases (with 1 layer corresponding to 2 extraction points). Due to the limited depth of the pretrained model, the gradient norm does not fully converge to zero. We also note a slight increase in the gradient norm near the final layers, which we attribute to interference from the softmax classifier at the end of the model, where the token to be generated is selected. This phenomenon was also observed in [1].
>
> **References**
>
> [1] Hieu M. Vu and Tan Minh Nguyen. Angular steering: Behavior control via rotation in activation space. NeurIPS, 2025.
>
> **Q4. Am I correct in understanding that the calculations for the steering functions (including momentum-based updates) are only when constructing the steering vectors -- when applying the steering vectors, the calculations stay the same as in ActAdd? Or are the you also changing the calculations done during inference?**
>
> **Answer:** Yes, that is correct. Our Momentum Steering method only modifies the construction of our steering vectors. During inference, we will apply the steering vectors in the exact same way as ActAdd or Directional Ablation, or the steering plane in Angular Steering, and thus will not introduce any new computational overhead.
>
> **Q5. Why choose Bregman divergence over other divergence functions?**
>
> **Answer:** We note that the Bregman Divergence is a generalization of many popular divergences, where choosing a suitable convex function $h$ in the Bregman Divergence recovers the desired divergence [1]. For example, if one chooses $h(x) = \frac{1}{2}||x||_2^2$ as we did in our manuscript, the Bregman Divergence becomes the (halved) squared Euclidean distance. With this choice of $h$, we are able to obtain a simple closed-form gradient that matches the steering updates. A full table of the different convex functions $h$ and the associated divergence can be found in [1], and modifying the choice of $h$ leads to new objectives and gradients, thus inducing new steering methods.
>
>
> **References**
>
> [1] Arindam Banerjee, Srujana Merugu, Inderjit S. Dhillon, and Joydeep Ghosh. 2005. Clustering with Bregman Divergences. J. Mach. Learn. Res. 6 (12/1/2005), 1705–1749.

---

> ### Author Response · Authors · 2025-11-19
> **Response to Reviewer VGaF (3)**
>
> **Q6. Can you provide intuition about how different choices of $h$ in the Bregman divergence lead to different feature maps and steering behaviors?**
>
> **Answer:** When $h$ is chosen to be quadratic, we recover the standard difference-in-means steering vector, as shown in our manuscript. However, for non-quadratic choices of $h$, different behaviors can emerge depending on the structure of $h$. For instance, if $h$ is designed to grow more rapidly for large-magnitude coordinates, the resulting gradients will place greater weight on those dimensions. More generally, by selecting $h$ that induces nonlinear transformations of individual features, the method can reshape the activation space$-$effectively stretching, compressing, or reweighting the steering direction along particular axes$-$yielding steering vectors with properties that differ fundamentally from the simple mean-difference approach.
>
>
> **Q7. How does this relate to matrix-based steering methods (Postmus & Abreu, 2025) or flow-based methods (Rodríguez et al, 2024)?**
>
> **Answer:** Matrix-based steering methods such as Conceptors [1] take a fundamentally different approach from vector-based steering. In [1], a conceptor matrix $C_l$ is computed per layer by solving an optimization problem using concept-related activations, and steering is performed via the layer-specific projection $h'_l = \beta_C C_l h_l$. Our method, by contrast, belongs to the vector-based steering literature, where steering directions are derived from contrasting datasets (typically via Difference-in-Means [2]) and aggregated across layers using a momentum mechanism to produce global directions. Thus, to the best of our knowledge, matrix-based conceptors and our momentum-aggregated steering vectors represent divergent methodological families.
>
> Flow-based methods such as [3] employ a causal formulation to compute steering parameters at each layer, which they refer to as the transport map $T$. While both [3] and our method account for the flow of activations and apply steering across all layers, [3] computes a separate transport map for each layer using the activations at that layer, whereas our method additionally aggregates the first moment across the model’s depth to compute the steering vectors. We also note that our method can be integrated into the Mean-AcT method proposed in [3]: by augmenting their transport map $T$ with our momentum-based procedure, we obtain Momentum-AcT, for which we report toxicity mitigation results in Table 4 of our manuscript.
>
> **References**
>
> [1] Joris Postmus, Steven Abreu. Steering Large Language Models using Conceptors: Improving Addition-Based Activation Engineering, Workshop: Foundation Model Interventions, NeurIPS, 2024.
>
> [2] Jan Wehner, Sahar Abdelnabi, Daniel Tan, David Krueger, Mario Fritz, "Taxonomy, Opportunities, and Challenges of Representation Engineering for Large Language Models", Preprint, 2025.
>
> [3] Pau Rodriguez, Arno Blaas, Michal Klein, Luca Zappella, Nicholas Apostoloff, marco cuturi, and Xavier Suau. Controlling language and diffusion models by transporting activations. ICLR, 2025.

---

> ### Author Response · Authors · 2025-11-19
> **Response to Reviewer VGaF (4)**
>
> **Q8. In equation 2.3, are you averaging over time? At which tokens do you extract activations and where do you apply them?**
>
>
> **Answer:** In Equation 2.3, we are averaging over the set of prompts, not over time. When extracting activations for computing the difference-in-means, we typically use the final token of each prompt, following the recommendations in [1], which is also the setup used in Section 4.1. However, for the experiments in Section 4.3, where we compare our method to [3], we follow their setup and use the activations of all tokens in the prompt when computing Equation 2.3. Finally, during inference, we apply steering at every generated token, as this has been shown to be more effective and is consistent with common practice in prior work [1, 2, 3, 4, 5].
>
> **References**
>
> [1] Hieu M. Vu and Tan Minh Nguyen. Angular steering: Behavior control via rotation in activation space. NeurIPS, 2025.
>
> [2] Jan Wehner, Sahar Abdelnabi, Daniel Tan, David Krueger, and Mario Fritz. Taxonomy, opportunities, and challenges of representation engineering for large language models, 2025.
>
> [3] Pau Rodriguez, Arno Blaas, Michal Klein, Luca Zappella, Nicholas Apostoloff, marco cuturi, and Xavier Suau. Controlling language and diffusion models by transporting activations. ICLR, 2025.
>
> [4] Andy Arditi, Oscar Balcells Obeso, Aaquib Syed, Daniel Paleka, Nina Rimsky, Wes Gurnee, and Neel Nanda. Refusal in language models is mediated by a single direction. NeurIPS, 2024.
>
> [5] Nishant Subramani, Nivedita Suresh, Matthew Peters.Extracting Latent Steering Vectors from Pretrained Language Models, Findings of ACL, 2022.
>
> -----
> We hope we have cleared your concerns about our work. We have also revised our manuscript according to your comments, and we would appreciate it if we could get your further feedback at your earliest convenience.

---

> ### Author Response · Authors · 2025-11-21
> **Regarding Q2 on Statistical Significance: Additional Results with Top-K Sampling**
>
> Dear Reviewer VGaF,
>
> We have completed additional experiments on the jailbreaking task using Top-K sampling rather than greedy decoding. For these experiments, we used Gemma2-9B-Instruct and evaluated two sampling settings: $K=10$ and $K=40$. Each configuration was run 10 times, and we report the summary statistics over these runs in Table 11 of Appendix I in our revised manuscript. For convenience, we also include the results in Table 1 below.
>
> Table 1: Performance of all configurations of our method with Gemma2-9B-Instruct on the jailbreaking task in Section 4.1, with Top-K ($K = 10, 40$ respectively) sampling over 10 runs. We report the mean and standard deviation of the ASR across the 10 runs for each configuration.
>
> |   Method  &nbsp; &nbsp;|&nbsp; &nbsp; &nbsp; Seq. &nbsp; &nbsp; &nbsp;|&nbsp; &nbsp; ASR $\uparrow$ &nbsp;|
> |:----------------|:-------------:|:-------------:|
> |Gemma2-9B-Instruct, $K = 10$|
> | AS (Baseline) |         |       $6.73\scriptstyle{\pm 1.11}$ |
> | AS + Mom. | | $39.71\scriptstyle{\pm 4.04}$ |
> | AS + Mom. (AA) | $\checkmark$| $40.29\scriptstyle{\pm 4.61}$ |
> | AS + Mom. (DA) | $\checkmark$| $37.69\scriptstyle{\pm 3.59}$ |
> | AS + Adam | | $29.52\scriptstyle{\pm 2.65}$|
> | AS + Adam (AA) | $\checkmark$| $34.04\scriptstyle{\pm 3.46}$ |
> | AS + Adam (DA) | $\checkmark$| $24.90\scriptstyle{\pm 2.65}$ |
> |Gemma2-9B-Instruct, $K = 40$|
> | AS (Baseline) |         |        $6.44\scriptstyle{\pm 0.65}$ |
> | AS + Mom. | | $38.94\scriptstyle{\pm 5.01}$ |
> | AS + Mom. (AA) | $\checkmark$| $40.96\scriptstyle{\pm 4.18}$ |
> | AS + Mom. (DA) | $\checkmark$| $37.79\scriptstyle{\pm 3.17}$ |
> | AS + Adam | | $29.13\scriptstyle{\pm 3.21}$ |
> | AS + Adam (AA) | $\checkmark$| $34.33\scriptstyle{\pm 4.23}$ |
> | AS + Adam (DA) | $\checkmark$| $25.29\scriptstyle{\pm 5.03}$ |

---

> ### Author Response · Authors · 2025-11-27
> **Reviewer VGaF Feedback Reminder**
>
> Dear Reviewer VGaF,
>
> Thank you once again for your thoughtful and constructive feedback. Your comments have been very helpful in improving the clarity and overall quality of our submission.
>
> We would like to kindly note that we uploaded our main rebuttal on November 18 (AoE), followed by additional experimental results on November 21 (AoE). With the discussion period ending soon, at 11:59 pm AoE on December 2, we want to ensure that there is adequate time to address any further questions or concerns you may have. After that point, reviewers will no longer be able to reply, and we will not be able to respond after 11:59 pm AoE on December 3.
>
> If there are any remaining issues you would like us to clarify, we would be grateful to hear from you while the discussion is still ongoing. We are happy to provide further explanations or to discuss any follow-up.
>
> If you feel that our responses have resolved the concerns raised in your initial review, we respectfully ask that you consider whether an updated score might better reflect your current assessment.
>
> Thank you again for your time, attention, and helpful insights.
>
> Sincerely,
>
> Authors

---

### Official Review · Reviewer_vAxe · 2025-11-02

**Soundness:** 3
**Presentation:** 3
**Contribution:** 2
**Rating:** 4
**Confidence:** 4

**Summary:**

This paper presents Momentum Steering, an optimization-inspired framework for activation steering in LLMs. In contrast with traditional methods, this method treats each layer independently, ignoring inter-layer dependencies and the temporal structure of representation formation. The authors formulate activation steering as a projected gradient descent (PGD) process that minimizes the Bregman divergence between activations for source and target prompts. From this optimization viewpoint, they introduce Momentum Steering, which incorporates momentum updates across layers to accumulate past feature directions

Two variants are proposed: Non-causal momentum steering, which aggregates difference-in-means statistics across layers. Causal momentum steering, which recursively updates future activations based on previous momentum directions, capturing causal dependencies.

Experimental results are reported across multiple benchmarks (e.g., ADVBENCH, RealToxicityPrompts, tinyBenchmarks) and model families (Gemma2, Qwen2.5, Llama3).

**Strengths:**

- Frames activation steering as a gradient-based optimization process, providing a clear theoretical foundation for a previously heuristic technique.

- The introduction of momentum accumulation across layers is conceptually interesting and well-motivated.

- The distinction between causal and non-causal formulations adds depth and flexibility, enabling adaptive control depending on task needs.

- Tests on a broad spectrum of models (3B–14B parameters) and tasks (toxicity mitigation, jailbreaks, and general benchmarks).

- Requires no retraining or fine-tuning, only modifying activation computations at inference time.

**Weaknesses:**

- While Appendix D mentions a stability analysis, it lacks detailed formal proofs or explicit convergence bounds for the momentum dynamics.

- Momentum accumulation, especially in causal sequential mode, increases computation during inference; efficiency metrics are not quantitatively reported.

- The link between optimization theory and steering behavior, might be an overkill for a traditionally cheap-compute task.

- The work focuses on quantitative results but lacks qualitative examples or visualization of how momentum steering alters model representations or outputs.

**Questions:**

Please see the weaknesses.

---

> ### Author Response · Authors · 2025-11-19
> **Response to Reviewer vAxe (1)**
>
> Thank you for your thoughtful review and valuable feedback. Below, we address your concerns.
>
> -----
>
>
> **Q1. While Appendix D mentions a stability analysis, it lacks detailed formal proofs or explicit convergence bounds for the momentum dynamics.**
>
> **Answer:** We appreciate the reviewer's comments on the stability analysis in Appendix E. In our revised manuscript, we have since refined our analysis to construct a discrete dynamical system that includes a time-varying bias. We believe that the updated derivation better coincides with the explicit choice of function $h$ in Section 3. From this, we provide a sufficient condition for the convergence of our method and its proof.
>
>
> **Q2. Momentum accumulation, especially in causal sequential mode, increases computation during inference; efficiency metrics are not quantitatively reported.**
>
> **Answer:**  Thank you for your suggestion. We would like to clarify that our method introduces no additional computational overhead during inference. Prior steering works (e.g., [1, 2, 3]) all require computing a feature direction along which the intervention is applied. Our momentum-based accumulation is an approach to this precomputation step, and it is executed once, offline, prior to any inference.
>
> Importantly, when integrated with our methods, the inference procedure in prior steering works remains unchanged: at runtime, they simply read the already-computed steering directions, exactly as they do in their original versions. As a result, our approach introduces no additional operations, memory usage, or latency on top of existing steering methods. All computational cost is confined to the offline estimation stage, not to inference.
>
> **References**
>
> [1] Andy Arditi, Oscar Balcells Obeso, Aaquib Syed, Daniel Paleka, Nina Rimsky, Wes Gurnee, and Neel Nanda. Refusal in language models is mediated by a single direction. NeurIPS, 2024.
>
> [2] Hieu M. Vu and Tan Minh Nguyen. Angular steering: Behavior control via rotation in activation space. NeurIPS, 2025.
>
> [3] Pau Rodriguez, Arno Blaas, Michal Klein, Luca Zappella, Nicholas Apostoloff, marco cuturi, and Xavier Suau. Controlling language and diffusion models by transporting activations. ICLR, 2025.
>
> **Q3. The link between optimization theory and steering behavior, might be an overkill for a traditionally cheap-compute task.**
>
> **Answer:** As explained in our answer to your Q2 above, our momentum-based steering methods do not introduce computational overhead at inference time.
>
> Also, please allow us to highlight the advantages of our optimization-based framework for steering LLM behavior.
>
> First, our approach offers a principled foundation for designing steering methods. By formulating steering as an optimization problem, as shown in Eqn. 17 of our manuscript, we can derive both existing and novel steering methods from first principles through the analysis of the objective function $J(x)$, the constraint set $C$, and the optimization algorithm used to solve Eqn. 17. In our work, we illustrate this framework by incorporating momentum-based acceleration techniques to develop momentum steering and Adam steering, which achieve substantial improvements over baseline steering methods across diverse tasks and LLM architectures (See Tables 1, 3, and 4 in our manuscript and revision).
>
> Second, optimization-based steering methods, such as our momentum steering and Adam steering, are particularly effective in challenging steering scenarios. While existing methods like angular steering perform well on simple tasks, they often struggle in more difficult settings, such as with Gemma2-9B-Instruct. As shown in Table 1 of our manuscript, both momentum and Adam steering successfully steer Gemma2-9B-Instruct.
>
> Finally, although not emphasized in the paper, our optimization-based framework also enables us to provide theoretical guarantees for steering methods by leveraging tools from optimization theory. In our work, we illustrate this capability by establishing convergence guarantees for our proposed momentum steering method.

---

> ### Author Response · Authors · 2025-11-19
> **Response to Reviewer vAxe (2)**
>
> **Q4. The work focuses on quantitative results but lacks qualitative examples or visualization of how momentum steering alters model representations or outputs.**
>
> **Answer:** Thank you for pointing out this area of improvement. We address this in Appendix F in the revised manuscript, where we compiled sample generations from Gemma2-9B-Instruct, steered using Angular Steering [1] with and without Momentum Steering, respectively, on a prompt from the test set used in our jailbreaking experiment in Section 4.1. The responses are compiled in Table 9. To summarize some key observations in the alteration of model outputs, when steering without Momentum Steering, we are unable to obtain an angle that can bypass the model's safety mechanisms. However, when steering with Momentum, we can both obtain a successful jailbreaking response and observe changes in behavior at different steering angles, as observed in [1].
>
> **References**
>
> [1] Hieu M. Vu and Tan Minh Nguyen. Angular steering: Behavior control via rotation in activation space. NeurIPS, 2025.
>
> -----
> We hope we have cleared your concerns about our work. We have also revised our manuscript according to your comments, and we would appreciate it if we could get your further feedback at your earliest convenience.

---

> ### Author Response · Authors · 2025-11-27
> **Reviewer vAxe Feedback Reminder**
>
> Dear Reviewer vAxe,
>
> Thank you once again for your thoughtful and constructive feedback. Your comments have been very helpful in improving the clarity and overall quality of our submission.
>
> We would like to kindly note that we uploaded our rebuttal on November 18 (AoE). With the discussion period ending soon, at 11:59 pm AoE on December 2, we want to ensure that there is adequate time to address any further questions or concerns you may have. After that point, reviewers will no longer be able to reply, and we will not be able to respond after 11:59 pm AoE on December 3.
>
> If there are any remaining issues you would like us to clarify, we would be grateful to hear from you while the discussion is still ongoing. We are happy to provide further explanations or to discuss any follow-up.
>
> If you feel that our responses have resolved the concerns raised in your initial review, we respectfully ask that you consider whether an updated score might better reflect your current assessment.
>
> Thank you again for your time, attention, and helpful insights.
>
> Sincerely,
>
> Authors

---

> > ### Comment · Reviewer_vAxe · 2025-11-27
> > **Response to authors**
> >
> > Thank you for the responses. While some of my concerns have been addressed I still find your story-line confusing:
> >
> > "This perspective connects steering vector construction to classical accelerated optimization, where momentum smooths trajectories and stabilizes convergence. We develop both non-causal and causal variants: the former aggregates difference-in-means statistics across layers, while the latter recursively incorporates the effect of previous interventions into future layer statistics."
> >
> > When you state the above highlighting core optimization concepts, one would expect that you are re-training the model. Is that correct? Then if this is the case, in some ways it defeats the original purpose of steering which is to control models without re-training them. If training is involved then one may as well just fine-tune the model on the target dataset.

---

> ### Author Response · Authors · 2025-11-27
>
> Thank you for your encouraging comments and further feedback.
>
> We would like to clarify that our method does not involve any re-training or fine-tuning of the model. This was also mentioned in our General Response to all of our reviewers, as well as in your review of the strengths of our paper, which had exactly the right idea.
>
> In prior steering methods such as ActAdd [1], Directional Ablation [2], Mean-AcT [3], or Angular Steering [4], we compute the difference-in-mean at each layer/extraction point. These can be utilized directly as steering vectors, as in ActAdd (although only one layer/extraction point is steered and this is chosen after hyperparameter tuning), or can be used as a collective to construct a steering plane, as in Angular Steering [4]. In our framework, these difference-in-means appear as the expected negative gradients (see Section 3.2 in our manuscript), and Momentum (or Adam) Steering simply proposes an accumulation of these negative gradients (or difference-in-means) to construct a richer set of steering vectors at each layer/extraction point. For example, in the non-sequential steering case with the difference-in-means computed at each layer/extraction point $k$, denoted as $r(k)$, our method constructs the steering vectors for each $k$ via the following simple update rule:
>
> $v(k) = \beta v(k-1) + r(k)$
>
> with $v(0)=0$. Furthermore, prior work has used momentum and optimization algorithms to design model architectures rather than to develop training procedures [5,6].
>
>
> In summary, we do not perform fine-tuning or model re-training, and Momentum Steering offers a way to aggregate the traditional difference-in-means to construct richer steering vectors via momentum updates. We hope this clarifies the concern you have with our method.
>
> **References**
>
> [1] Alexander Matt Turner, Lisa Thiergart, Gavin Leech, David Udell, Juan J Vazquez, Ulisse Mini, Monte MacDiarmid. Steering Language Models with Activation Engineering. OpenReview preprint, 2023.
>
> [2] Andy Arditi, Oscar Balcells Obeso, Aaquib Syed, Daniel Paleka, Nina Rimsky, Wes Gurnee, and Neel Nanda. Refusal in language models is mediated by a single direction. NeurIPS, 2024.
>
> [3] Pau Rodriguez, Arno Blaas, Michal Klein, Luca Zappella, Nicholas Apostoloff, marco cuturi, and Xavier Suau. Controlling language and diffusion models by transporting activations. ICLR, 2025.
>
> [4] Hieu M. Vu and Tan Minh Nguyen. Angular steering: Behavior control via rotation in activation space. NeurIPS, 2025.
>
> [5] Michael E Sander, Pierre Ablin, Mathieu Blondel, Gabriel Peyré. Momentum residual neural networks. PMLR, 2021.
>
> [6] Rachel Teo and Tan Minh Nguyen. MomentumSMoe: Integrating momentum into sparse mixture of experts. NeurIPS, 2024.

---

### Official Review · Reviewer_xjsN · 2025-11-10

**Soundness:** 3
**Presentation:** 2
**Contribution:** 3
**Rating:** 4
**Confidence:** 2

**Summary:**

The paper introduces Momentum Steering, a new framework for activation steering in LLMs. Activation steering refers to the manipulation of hidden activations at inference time (e.g., with methods like Activation Addition or Directional Ablation) to control model behavior without retraining.

The authors propose viewing traditional steering methods (based on difference-in-means activations between contrastive prompts) as instances of an optimization problem solvable via projected gradient descent ((P)GD), for which they provide a proof.

Based on this insight, they incorporate momentum into the process of constructing steering vectors, enabling the accumulation of information across layers. They further generalize the idea to Adam Steering, which applies Adam-style updates to steering vectors.

Empirical results across mutliple LLMs (Qwen2.5, Llama3, Gemma2) and tasks (jailbreaking task, toxicity reduction, and general language performance) show that Momentum and Adam Steering produce stronger behavioral control, while maintaining general model performance.

**Strengths:**

- **Comprehensive evaluation**: The authors test across a wide variety of models and multiple tasks, which supports the generality of their claims.
- **Empirically strong results**: Momentum Steering consistently improves behavioral control, while maintaing general language performance. Given the importance of LLM alignment problems, this seems like a significant result.
- **Conceptual novelty**: Reducing activation steering to a gradient descent-based optimization problem, thereby allowing for the usage of momentum mechanism, is an oroginal and interesting idea.

**Weaknesses:**

- **Missing citation**: Please provide a citation for the claim in line 53, as it is central to your paper: "This design can overlook valuable structure across layers, producing unstable or underpowered feature directions, especially in deeper models or tasks requiring fine-grained control."
- **Clarity of the (P)GD proof**: The proof that activation steering methods reduce to (projected) gradient descent updates should be clearer. It's a bit difficult to follow in its current state.
	- Notation for equations 1 and 2 is not consistent with equation in line 104: x(k, p) vs x(k)(p)
	- The "Sequential Refinements" section lacks clarity. It's not evident from the equations how Mean-AcT introduces feedback and the change in dataset notation (D^{(train)}) in comparison to line 105 is not explained.
	- Lines 150-152: the variable $p$ is used for prompts from different datasets, which can cause confusion. Additionally, introducing symmetric notation (e.g., x_{src} and x_{tg}) would improve readability.
	- Line 184: The introduction of the convex constraint set into the optimization in Eqn. 9 is non-trivial and deserves more intuition or justification.
	- Line 198: The hypothesis that the layer function $f(k)$ implicitly performs the projection $P_C$ is non-trivial and unproven. Since this step appears to be an important element in the reduction to activation steering, additional theoretical or empirical support would be important.
- **Experimental baseline**: The paper should more explicitly define what the baseline is in the jailbreaking experiment. Is it standard activation steering (e.g. based on Turner et al. 2023) or the (P)GD-based approach? Direct comparisons to existing and established methods that do not rely on the optimization framing would make the empirical results more interpretable (similar to what you do in section 4.3).
- **Minor issues and typos**
	- Typo in line 112 "Meant-AcT" --> "Mean-AcT"
	- Line 125: should read "uses" or "incorporates"
	- Line 363: should read "might require significantly more time"

**Questions:**

I discuss this primarily in the weaknesses section, but the proof that activation steering methods reduce to (projected) gradient descent updates is not yet entirely convincing. Addressing the points mentioned above would help clarify this argument, particularly the assumption that the layer function implicitly performs the projection step.

In addition, the “Empirical Evidence” section could be expanded to strengthen this claim. For example, showing that the steering vectors derived from your optimization-based reduction are sufficiently similar to those obtained from existing methods such as ActAdd (e.g., via geometrical or downstream performance comparisons) would make the theoretical correspondence much more compelling.

---

> ### Author Response · Authors · 2025-11-19
> **Response to Reviewer xjsN (1)**
>
> Thank you for your thoughtful review and valuable feedback. Below, we address your concerns.
>
> -----
>
> **Q1. Missing citation:**
> **Missing citation: Please provide a citation for the claim in line 53, as it is central to your paper: "This design can overlook valuable structure across layers, producing unstable or underpowered feature directions, especially in deeper models or tasks requiring fine-grained control."**
>
>
> **Answer:** We thank the reviewer for pointing out the missing citation. We have updated the section to include the citation as follows:
>
> Sequential extensions [7] refine this idea by propagating steering parameters estimation layer by layer, but the underlying update remains memoryless; that is, each layer has its own steering transformation. This design can overlook valuable structure across layers, leading to unstable or underpowered feature directions, especially in deeper models or tasks that require fine-grained control. Specifically, prior work has shown that layers in LLMs exhibit substantial coupling [1, 2, 3], implying that their representation spaces share a coherent global structure. Recent studies further demonstrate that interventions and analyses applied across multiple layers, rather than a single layer, produce more reliable effects [4, 5, 6], underscoring the importance of inter-layer dependencies for effective steering.
>
> **References**
>
> [1] Kevin Ro Wang, Alexandre Variengien, Arthur Conmy, Buck Shlegeris, and Jacob Steinhardt. Interpretability in the wild: a circuit for indirect object identification in GPT-2 small. ICLR, 2023.
>
> [2] Thomas McGrath, Matthew Rahtz, Janos Kramar, Vladimir Mikulik, and Shane Legg. The hydra effect: Emergent self-repair in language model computations. Deepmind, 2023.
>
> [3] Cody Rushing and Neel Nanda. Explorations of self-repair in language models. ICML, 2024.
>
> [4] Jack Lindsey, Adly Templeton, Jonathan Marcus, Thomas Conerly, Joshua Batson, and Christopher Olah. Sparse crosscoders for cross-layer features and model diffing. Transformer Circuits Thread, 2024.
>
> [5] Andy Arditi, Oscar Balcells Obeso, Aaquib Syed, Daniel Paleka, Nina Rimsky, Wes Gurnee, and Neel Nanda. Refusal in language models is mediated by a single direction. NeurIPS, 2024.
>
> [6] Hieu M. Vu and Tan Minh Nguyen. Angular steering: Behavior control via rotation in activation space. NeurIPS, 2025.
>
> [7] Pau Rodriguez, Arno Blaas, Michal Klein, Luca Zappella, Nicholas Apostoloff, Marco Cuturi, Xavier Suau, Controlling Language and Diffusion Models by Transporting Activations, ICLR, 2025.

---

> ### Author Response · Authors · 2025-11-19
> **Response to Reviewer xjsN (2)**
>
> **Q2. Clarity of the (P)GD proof: The proof that activation steering methods reduce to (projected) gradient descent updates should be clearer. It's a bit difficult to follow in its current state.**
>
> **Notation for equations 1 and 2 is not consistent with equation in line 104: x(k, p) vs x(k)(p)**
>
> **The "Sequential Refinements" section lacks clarity. It's not evident from the equations how Mean-AcT introduces feedback and the change in dataset notation (D^{(train)}) in comparison to line 105 is not explained.**
>
> **Lines 150-152: the variable $p$ is used for prompts from different datasets, which can cause confusion. Additionally, introducing symmetric notation (e.g., x_{src} and x_{tg}) would improve readability.**
>
> **Line 184: The introduction of the convex constraint set into the optimization in Eqn. 9 is non-trivial and deserves more intuition or justification.**
>
> **Line 198: The hypothesis that the layer function $f(k)$ implicitly performs the projection $P_{C}$ is non-trivial and unproven. Since this step appears to be an important element in the reduction to activation steering, additional theoretical or empirical support would be important.**
>
> **But the proof that activation steering methods reduce to (projected) gradient descent updates is not yet entirely convincing. Addressing the points mentioned above would help clarify this argument, particularly the assumption that the layer function implicitly performs the projection step.**
>
>  **Answer:** Thanks for pointing out these confusions. We have revised our manuscript to address them. Please allow us to clarify our GD proof and reply to your concerns below.
>
>  **Notation for equations 1 and 2 is not consistent with equation in line 104: $x(k, p)$ vs. $x(k)(p)$**
>
>  We have fixed this typo in our revision.
>
>  **The "Sequential Refinements" section lacks clarity. It's not evident from the equations how Mean-AcT introduces feedback, and the change in dataset notation (D^{(train)}) in comparison to line 105 is not explained.**
>
> We agree with the reviewer that describing Mean-AcT as introducing "feedback" in the Sequential Refinements section was vague. By "feedback", we just meant that current-layer interventions in Mean-Act account for earlier interventions, even though the steering vectors are still computed independently at each layer. We have removed the term feedback and added this clarification to the Sequential Refinements section in the revision.
>
>
> In our revision, we have also defined and clarified $D^{\text{(train)}}\_{\text{source}}$ and $D^{\text{(train)}}\_{\text{target}}$ from line 107 to line 116 in our revised manuscript. In particular, we explain that $D^{\text{(train)}}_{\text{source}}$ and $D^{\text{(train)}}\_{\text{target}}$ are used to compute the steering vectors $r{(k)}$. These are different from $D\_{\text{source}}$ and $D\_{\text{target}}$ in Eqns. 1 and 2, which contain the prompts at inference time.
>
> **Lines 150-152: the variable $p$ is used for prompts from different datasets, which can cause confusion. Additionally, introducing symmetric notation (e.g., x_{src} and x_{tg}) would improve readability.**
>
>  In our revised manuscript, we have used $p\_{tg}$ for the input prompts from the target data $\mathcal{D}\_{\text{target}}$, i.e., $p\_{tg} \in \mathcal{D}\_{\text{target}}$. We also agree with the reviewer that introducing symmetric notation (e.g., $x_{src}$ and $x_{tg}$) would help improve readability. However, for notational simplicity, we still use $x(t,p)$, instead of $x_{src}(t,p_{src})$, to denote the activation at time $t$ when processing the input prompt $p \in \mathcal{D}_{\text{source}}$. We have explained this choice of notations from line 162 to line 164 in our revision.

---

> ### Author Response · Authors · 2025-11-19
> **Response to Reviewer xjsN (3)**
>
> **Line 184: The introduction of the convex constraint set into the optimization in Eqn. 9 is non-trivial and deserves more intuition or justification.**
>
> We have added further explanation for the introduction of the convex constraint set into the optimization in Eqn. 9 from line 200 to line 205 in our revision. In particular, the convex constraint sets result from certain properties of the activation $x(t)$. For example, the activations $x(t)$ are (lower) bounded due to the activation functions such as ReLU or SwiGLU, or the norms of $x(t)$ are bounded due to the $\mathrm{Norm}$ operators (see Section 2.1 in our manuscript and revision) such as layer normalization (LayerNorm) or Root Mean Square normalization (RMSNorm). These properties define convex constraint sets on $x(t)$. The low-rank properties of $x(t)$ mentioned in our manuscript do not result in a convex constraint set, but a non-convex constraint set. However, projected gradient descent (PGD), discussed in the "How about the layer function f (k) in Eqn. 2?" in Section 3.2 of our manuscript, can still be used to solve the corresponding non-convex constrained optimization problem with convergence guarantees under certain conditions [1]. We have discussed this in Remark 4, from lines 252 to 257, of our revision.
>
>  **Line 198: The hypothesis that the layer function $f(k)$ implicitly performs the projection $P_{C}$ is non-trivial and unproven. Since this step appears to be an important element in the reduction to activation steering, additional theoretical or empirical support would be important.**
>
>  The projection $P_C$ is, in fact, induced by the layer function $f(k)$. Specifically, the convex constraint sets on $x(t)$ arise from the components within $f(k)$. For example, as mentioned above, the ReLU and SwiGLU activation functions in $f(k)$ induce the bounded properties of $x(t)$ while the LayerNorm and RMSNorm operators induce the norm-bounded properties of $x(t)$. Thus, it is not a hypothesis anymore, but these convex constraints are a direct consequence of the layer function $f(k)$, which effectively acts as a projection onto the corresponding convex sets. We have incorporated this clarification into the section "How about the layer function $f(k)$ in Eqn. 2?" in Section 3.2 of our revision.
>
> **References**
>
> [1] Rina Foygel Barber and Wooseok Ha. Gradient descent with non-convex constraints: local concavity determines convergence. Information and Inference: A Journal of the IMA, 7(4):755–806, 2018.
>
> **Q3: Experimental baseline:  The paper should more explicitly define what the baseline is in the jailbreaking experiment. Is it standard activation steering (e.g. based on Turner et al. 2023) or the (P)GD-based approach? Direct comparisons to existing and established methods that do not rely on the optimization framing would make the empirical results more interpretable (similar to what you do in section 4.3).**
>
>  **Answer:** Thank you for your suggestion. We would like to clarify that the baseline of the jailbreaking experiment in Section 4.1 (and Section 4.2) refers to using regular Angular Steering with difference-in-means as proposed in [1]. This can also be found in Line 317-319 of the original manuscript or Line 365-367 in the revised manuscript. In that experiment, we demonstrate that using Momentum (or Adam) Steering to construct the steering vectors/plane yields better results in the jailbreaking task than using the regular difference-in-means.
>
> We would also like to note that methods such as ActAdd, Directional Ablation, and Angular Steering only utilize the steering vector or steering plane during inference [1,2,3], whereas our method primarily modifies the construction of the steering vector rather than the regular difference-in-mean. Thus, our method is complementary and can be directly applied on top of these methods to enhance their performance, even though the existing methods might not rely on our optimization framework.
>
> As an additional example, we have implemented our method using ActAdd and have included the additional results in Table 10 in Appendix H of our revision.
>
> **References**
>
> [1] Hieu M. Vu and Tan Minh Nguyen. Angular steering: Behavior control via rotation in activation space. NeurIPS, 2025.
>
> [2] Andy Arditi, Oscar Balcells Obeso, Aaquib Syed, Daniel Paleka, Nina Rimsky, Wes Gurnee, and Neel Nanda. Refusal in language models is mediated by a single direction. NeurIPS, 2024.
>
> [3] Nina Rimsky, Nick Gabrieli, Julian Schulz, Meg Tong, Evan Hubinger, and Alexander Turner. Steering llama 2 via contrastive activation addition. ACL, 2024.

---

> ### Author Response · Authors · 2025-11-19
> **Response to Reviewer xjsN (4)**
>
> **Q4: Minor issues and typos**
>   **- Typo in line 112: *Mean-AcT* → *MeanAct*.**
>
>   **- Line 125: "uses" → "incorporates."**
>
>    **- Line 363: should read "might require significantly more time"**
>
>  **Answer:** Thank you for pointing out these typos. We have fixed them in our revised manuscript. However, we would like to note that "Mean-AcT" isn't a typo, as we are following the name used in [1].
>
>  **References**
>
> [1] Pau Rodriguez, Arno Blaas, Michal Klein, Luca Zappella, Nicholas Apostoloff, marco cuturi, and Xavier Suau. Controlling language and diffusion models by transporting activations. ICLR, 2025.
>
> **Q5. [Regarding the proof that activation steering methods reduce to (projected) gradient descent updates] In addition, the "Empirical Evidence" section could be expanded to strengthen this claim. For example, showing that the steering vectors derived from your optimization-based reduction are sufficiently similar to those obtained from existing methods such as ActAdd (e.g., via geometrical or downstream performance comparisons) would make the theoretical correspondence much more compelling.**
>
>  **Answer:** Thank you for the suggestion. We would first like to note that by setting the momentum coefficient, $\beta = 0$, the steering vectors in each layer reduce to the regular difference-in-means, which are used in methods such as ActAdd, Directional Ablation, or Angular Steering. As mentioned in our response to Q3, we have compared ActAdd with and without Momentum Steering in Table 10, Appendix H, in our revision. Furthermore, to supplement the jailbreaking experiment in Section 4.1 (as well as Appendix C), we also perform an additional ablation study on the effect of the momentum coefficient $\beta$ when using Momentum Steering on top of regular Angular Steering. The results for this are added in Table 5, Section 4.4, and Table 8, Appendix D, in the revised manuscript.
>
> -----
> We hope we have cleared your concerns about our work. We have also revised our manuscript according to your comments, and we would appreciate it if we could get your further feedback at your earliest convenience.

---

> ### Author Response · Authors · 2025-11-27
> **Reviewer xjsN Feedback Reminder**
>
> Dear Reviewer xjsN,
>
> Thank you once again for your thoughtful and constructive feedback. Your comments have been very helpful in improving the clarity and overall quality of our submission.
>
> We would like to kindly note that we uploaded our rebuttal on November 18 (AoE). With the discussion period ending soon, at 11:59 pm AoE on December 2, we want to ensure that there is adequate time to address any further questions or concerns you may have. After that point, reviewers will no longer be able to reply, and we will not be able to respond after 11:59 pm AoE on December 3.
>
> If there are any remaining issues you would like us to clarify, we would be grateful to hear from you while the discussion is still ongoing. We are happy to provide further explanations or to discuss any follow-up.
>
> If you feel that our responses have resolved the concerns raised in your initial review, we respectfully ask that you consider whether an updated score might better reflect your current assessment.
>
> Thank you again for your time, attention, and helpful insights.
>
> Sincerely,
>
> Authors

---

### Author Response · Authors · 2025-11-19
**Summary of Revision**

After integrating the suggestions provided from all reviewers, we summarize the revisions made to our manuscript below:

1. We have updated the section containing the missing citations to include the relevant citations.
2. We have revised Sections 2 and 3 to improve the presentation and clarity of our notation, background, and theory related to our optimization framework.
3. We have added an additional ablation study on the role of the momentum coefficient, $\beta$, on Gemma2-9B-Instruct and Gemma2-27B-Instruct performed under the setting of the jailbreaking experiment. The results for this are presented in Tables 5 and 8 of Section 4.4 and Appendix D, respectively, in our revised manuscript.
4. We have revised our Stability Analysis in Appendix E (previously Appendix D in our original manuscript) such that the discrete dynamical system agrees with the choice of $h$ in Section 3. We have also included a sufficient condition for convergence along with the corresponding proofs in the revised manuscript.
5. We have compiled a set of responses in Table 9, Appendix F, generated from Gemma2-9B-Instruct, on a prompt used from the test set of our jailbreaking experiment. The table shows the different responses the model gives when there is no steering, steering using Angular Steering without Momentum Steering, or steering using Angular Steering with Momentum Steering.
6. We have added plots of the norms of the steering vectors (computed sequentially) from the pretrained models Llama3.2-3B-Instruct and Llama3.1-8B-Instruct in Figures 4 and 5, respectively, under Appendix G.
7. We have conducted an additional experiment on the jailbreaking task using ActAdd instead of Angular Steering as our method of intervention during inference, and have reported the results in Table 10, Appendix H, in our revised manuscript.
8. We have conducted an additional experiment on the jailbreaking task but using Top-K sampling (with $K = 10, 40$ respectively) during inference instead of greedy decoding as performed in Section 4.1. We report the mean and standard deviation of our results in Table 11, Appendix I, in our revised manuscript.
9. We have fixed the typos pointed out in the reviews and added minor revisions across the manuscript for better clarity.

---

---

### Author Response · Authors · 2025-11-20
**General Response (1)**

Dear AC and Reviewers,

Thank you for your thoughtful reviews and valuable feedback, which have greatly strengthened our paper. We are encouraged by the positive assessments noting that: **(1)** framing activation steering as a gradient descent-based optimization problem, thereby enabling the use of momentum mechanisms, is an original and interesting idea (Reviewer xjsN, vAxe, VGaF); **(2)** our framework provides a clear theoretical foundation for a previously heuristic steering technique (Reviewer vAxe), and our proposed momentum steering is supported by stability analysis and connects to well-established optimization literature (Reviewer VGaF); and **(3)** our momentum-based steering methods yield consistent improvements across a broad range of models and tasks, are easy to incorporate into existing steering approaches, and require neither retraining nor significant computational overhead (Reviewers xjsN, vAxe, VGaF). We have revised the submission in response to all reviewer comments, and **all changes are highlighted in magenta**.

**One of the main concerns raised by the reviewers is that Momentum accumulation, especially in causal sequential mode, increases computation during inference.**

To address this, we would like to clarify that our method introduces no additional computational overhead during inference. Prior steering works (e.g., [1, 2, 3]) all require computing a feature direction along which the intervention is applied. Our momentum-based accumulation is an approach to this precomputation step, and it is executed once, offline, prior to any inference.

Importantly, when integrated with our methods, the inference procedure in prior steering works remains unchanged: at runtime, they simply read the already-computed steering directions, exactly as they do in their original versions. As a result, our approach introduces no additional operations, memory usage, or latency on top of existing steering methods. All computational cost is confined to the offline estimation stage, not to inference.

---

> ### Author Response · Authors · 2025-11-20
> **General Response (2)**
>
> **Another concern is that the argument showing how activation steering methods reduce to (projected) gradient descent updates was not sufficiently clear.**
>
> We address key questions from reviewers regarding our derivation below.
>
> >The introduction of the convex constraint set into the optimization in Eqn. 9
>
> We have added further explanation for the introduction of the convex constraint set into the optimization in Eqn. 9 from line 200 to line 205 in our revision. In particular, the convex constraint sets result from certain properties of the activation $x(t)$. For example, the activations $x(t)$ are (lower) bounded due to the activation functions such as ReLU or SwiGLU, or the norms of $x(t)$ are bounded due to the $\mathrm{Norm}$ operators (see Section 2.1 in our manuscript and revision) such as layer normalization (LayerNorm) or Root Mean Square normalization (RMSNorm). These properties define convex constraint sets on $x(t)$. The low-rank properties of $x(t)$ mentioned in our manuscript do not result in a convex constraint set, but a non-convex constraint set. However, projected gradient descent (PGD), discussed in the "How about the layer function f (k) in Eqn. 2?" in Section 3.2 of our manuscript, can still be used to solve the corresponding non-convex constrained optimization problem with convergence guarantees under certain conditions [1]. We have discussed this in Remark 4, from lines 252 to 257, of our revision.
>
> >The hypothesis that the layer function $f(k)$ implicitly performs the projection $P_{C}$
>
>  The projection $P_C$ is, in fact, induced by the layer function $f(k)$. Specifically, the convex constraint sets on $x(t)$ arise from the components within $f(k)$. For example, as mentioned above, the ReLU and SwiGLU activation functions in $f(k)$ induce the bounded properties of $x(t)$ while the LayerNorm and RMSNorm operators induce the norm-bounded properties of $x(t)$. Thus, it is not a hypothesis anymore, but these convex constraints are a direct consequence of the layer function $f(k)$, which effectively acts as a projection onto the corresponding convex sets. We have incorporated this clarification into the section "How about the layer function $f(k)$ in Eqn. 2?" in Section 3.2 of our revision.
>
> > Bregman divergence over other divergence functions?
>
> We note that the Bregman Divergence is a generalization of many popular divergences, where choosing a suitable convex function $h$ in the Bregman Divergence recovers the desired divergence [4]. For example, if one chooses $h(x) = \frac{1}{2}||x||_2^2$ as we did in our manuscript, the Bregman Divergence becomes the (halved) squared Euclidean distance. With this choice of $h$, we are able to obtain a simple closed-form gradient that matches the steering updates. A full table of the different convex functions $h$ and the associated divergence can be found in [4], and modifying the choice of $h$ leads to new objectives and gradients, thus inducing new steering methods.
>
> > Intuition about how different choices of $h$ in the Bregman divergence lead to different feature maps and steering behaviors
>
> When $h$ is chosen to be quadratic, we recover the standard difference-in-means steering vector, as shown in our manuscript. However, for non-quadratic choices of $h$, different behaviors can emerge depending on the structure of $h$. For instance, if $h$ is designed to grow more rapidly for large-magnitude coordinates, the resulting gradients will place greater weight on those dimensions. More generally, by selecting $h$ that induces nonlinear transformations of individual features, the method can reshape the activation space$-$effectively stretching, compressing, or reweighting the steering direction along particular axes$-$yielding steering vectors with properties that differ fundamentally from the simple mean-difference approach.
>
> **References**
>
> [1] Andy Arditi, Oscar Balcells Obeso, Aaquib Syed, Daniel Paleka, Nina Rimsky, Wes Gurnee, and Neel Nanda. Refusal in language models is mediated by a single direction. NeurIPS, 2024.
>
> [2] Hieu M. Vu and Tan Minh Nguyen. Angular steering: Behavior control via rotation in activation space. NeurIPS, 2025.
>
> [3] Pau Rodriguez, Arno Blaas, Michal Klein, Luca Zappella, Nicholas Apostoloff, marco cuturi, and Xavier Suau. Controlling language and diffusion models by transporting activations. ICLR, 2025.
>
> [4] Arindam Banerjee, Srujana Merugu, Inderjit S. Dhillon, and Joydeep Ghosh. Clustering with Bregman Divergences. JMLR, 2005.
>
>
>
> ----
>
> We are glad to answer any further questions you have on our submission.
>
> ----

---

### Author Response · Authors · 2025-11-24
**Follow-Up After Rebuttal Submission**

Dear Reviewers and Chairs,

Thank you very much for your insightful comments and feedback, and for the time you have invested in reviewing our work. We also appreciate the significant effort the chairs put into coordinating the process and facilitating a constructive exchange.

We have now finalized and uploaded our updated rebuttal. All planned revisions and additional results have been incorporated into the document and are summarized in the “Summary of Revision” section.

We would greatly appreciate hearing whether our explanations and new experiments sufficiently address your concerns, or if there are any remaining questions we should clarify.

We are happy to continue the discussion and provide any additional information that may be useful.

Warm regards,

Authors

---

### Author Response · Authors · 2025-12-01
**Additional Results on Gemma2-27B-Instruct for the Jailbreaking Task**

Dear Chairs,

We have completed additional experiments on the Gemma2-27B-Instruct model with respect to the jailbreaking task. In Table 6, Appendix C of our original manuscript, we note that the performance of Adam Steering on the Gemma2-27B-Instruct model was suboptimal. However, upon changing the original moment coefficients used in the experiments ($\beta_1=0.9$, $\beta_2=0.999$) to a new set of coefficients ($\beta_1=0.999$, $\beta_2=0.5$), we were able to observe significant performance gains as compared to baseline. The new results have been updated in Table 7, Appendix C, highlighted in Magenta in the revised manuscript.

Additionaly, we have also ran the Top-K sampling experiments ($K=10,40$ respectively), with identical settings described in the response to Reviewer VGaF as well as in Appendix I of the revised manuscript. The results have been appended in Table 11, Appendix I, and we have also included them here in Table 1 for convenience. We note that the results of the Top-K sampling experiments (for both Gemma2-9B-Instruct and Gemma2-27B-Instruct) indicate that even under non-greedy decoding settings, Momentum and Adam Steering continue to yield stronger steering effects.

Table 1: Performance of all configurations of our method with Gemma2-27B-Instruct on the jailbreaking task in Section 4.1, with Top-K ($K = 10, 40$ respectively) sampling over 10 runs. We report the mean and standard deviation of the ASR across the 10 runs for each configuration.

|   Method  &nbsp; &nbsp;|&nbsp; &nbsp; &nbsp; Seq. &nbsp; &nbsp; &nbsp;|&nbsp; &nbsp; ASR $\uparrow$ &nbsp;|
|:----------------|:-------------:|:-------------:|
|Gemma2-27B-Instruct, $K = 10$|
| AS (Baseline) |         |       $4.42\scriptstyle{\pm 1.22}$ |
| AS + Mom. | | $67.40\scriptstyle{\pm 2.74}$ |
| AS + Mom. (AA) | $\checkmark$| $49.81\scriptstyle{\pm 4.22}$ |
| AS + Mom. (DA) | $\checkmark$| $43.46\scriptstyle{\pm 3.39}$ |
| AS + Adam | | $49.81\scriptstyle{\pm 3.29}$|
| AS + Adam (AA) | $\checkmark$| $26.83\scriptstyle{\pm 3.49}$ |
| AS + Adam (DA) | $\checkmark$| $31.35\scriptstyle{\pm 3.27}$ |
|Gemma2-27B-Instruct, $K = 40$|
| AS (Baseline) |         |        $4.81\scriptstyle{\pm 1.28}$ |
| AS + Mom. | | $67.50\scriptstyle{\pm3.32}$ |
| AS + Mom. (AA) | $\checkmark$| $49.52\scriptstyle{\pm 3.02}$ |
| AS + Mom. (DA) | $\checkmark$| $44.13\scriptstyle{\pm 3.25}$ |
| AS + Adam | | $51.63\scriptstyle{\pm 2.72}$ |
| AS + Adam (AA) | $\checkmark$| $26.63\scriptstyle{\pm 3.51}$ |
| AS + Adam (DA) | $\checkmark$| $31.63\scriptstyle{\pm 2.29}$ |

---

### Author Response · Authors · 2025-12-01
**To New AC - Summary of Novelties and Contributions of Our Work**

Dear new AC,

Thank you for assuming the handling of our submission. To assist in the understanding of our work, we summarize the novelties and contributions of our paper as follows:

1. We provide a theoretical framework that interprets activation steering as a (projected) gradient descept step that aims at minimizing an underlying objective function. In particular, the traditional difference-in-means at each layer/extraction point, normally used as steering vectors in prior steering methods such as ActAdd [1] and Mean-AcT [2], can be observed as expected negative gradients with respect to our optimization framework.  This is summarized by Theorem 1, Section 3.2, in both our original and revised manuscript.
2. We propose Momentum (and Adam) Steering in Section 3.3 (and Section 3.4 respectively) that accumulates the raw difference-in-means via a momentum buffer (or first and second moment buffer in Adam Steering) to construct a richer set of steering vectors for each layer/extraction point. This can be done either sequentially or non-sequentially (Definition 1 and 2 in Section 3.3 and Section 3.4 respectively).
3. We test our methods on the jailbreaking (Section 4.1, 4.2, Appendix C) and toxicity mitigation task (Section 4.3) and across a wide array of model families (Qwen2.5, Llama3, Gemma2 with models ranging from 3-14B parameters in Section 4 and larger models of 27B or 32B parameters in Appendix C). The results show that our methods were able to significantly improve the steering effect across multiple models under different task settings.


**References**

[1] Alexander Matt Turner, Lisa Thiergart, Gavin Leech, David Udell, Juan J Vazquez, Ulisse Mini, Monte MacDiarmid. Steering Language Models with Activation Engineering. OpenReview preprint, 2023.

[2] Pau Rodriguez, Arno Blaas, Michal Klein, Luca Zappella, Nicholas Apostoloff, marco cuturi, and Xavier Suau. Controlling language and diffusion models by transporting activations. ICLR, 2025.

---

### Author Response · Authors · 2025-12-01
**To New AC - Summary of Primary Reviewer Questions and Our Responses (1)**

Dear new AC,

In this message, we would like to provide a summary of the key concerns highlighted by our reviewers in their initial reviews as well as our responses that addresses them. This can be found in our General Response towards all the reviewers but we shall list them here for convenience:

**One of the main concerns raised by the reviewers is that Momentum accumulation, especially in causal sequential mode, increases computation during inference.**

To address this, we would like to clarify that our method introduces no additional computational overhead during inference. Prior steering works (e.g., [1, 2, 3]) all require computing a feature direction along which the intervention is applied. Our momentum-based accumulation is an approach to this precomputation step, and it is executed once, offline, prior to any inference.

Importantly, when integrated with our methods, the inference procedure in prior steering works remains unchanged: at runtime, they simply read the already-computed steering directions, exactly as they do in their original versions. As a result, our approach introduces no additional operations, memory usage, or latency on top of existing steering methods. All computational cost is confined to the offline estimation stage, not to inference.

---

> ### Author Response · Authors · 2025-12-01
> **To New AC - Summary of Primary Reviewer Questions and Our Responses (2)**
>
> **Another concern is that the argument showing how activation steering methods reduce to (projected) gradient descent updates was not sufficiently clear.**
>
> We address key questions from reviewers regarding our derivation below.
>
> >The introduction of the convex constraint set into the optimization in Eqn. 9
>
> We have added further explanation for the introduction of the convex constraint set into the optimization in Eqn. 9 from line 200 to line 205 in our revision. In particular, the convex constraint sets result from certain properties of the activation $x(t)$. For example, the activations $x(t)$ are (lower) bounded due to the activation functions such as ReLU or SwiGLU, or the norms of $x(t)$ are bounded due to the $\mathrm{Norm}$ operators (see Section 2.1 in our manuscript and revision) such as layer normalization (LayerNorm) or Root Mean Square normalization (RMSNorm). These properties define convex constraint sets on $x(t)$. The low-rank properties of $x(t)$ mentioned in our manuscript do not result in a convex constraint set, but a non-convex constraint set. However, projected gradient descent (PGD), discussed in the "How about the layer function f (k) in Eqn. 2?" in Section 3.2 of our manuscript, can still be used to solve the corresponding non-convex constrained optimization problem with convergence guarantees under certain conditions [1]. We have discussed this in Remark 4, from lines 252 to 257, of our revision.
>
> >The hypothesis that the layer function $f(k)$ implicitly performs the projection $P_{C}$
>
>  The projection $P_C$ is, in fact, induced by the layer function $f(k)$. Specifically, the convex constraint sets on $x(t)$ arise from the components within $f(k)$. For example, as mentioned above, the ReLU and SwiGLU activation functions in $f(k)$ induce the bounded properties of $x(t)$ while the LayerNorm and RMSNorm operators induce the norm-bounded properties of $x(t)$. Thus, it is not a hypothesis anymore, but these convex constraints are a direct consequence of the layer function $f(k)$, which effectively acts as a projection onto the corresponding convex sets. We have incorporated this clarification into the section "How about the layer function $f(k)$ in Eqn. 2?" in Section 3.2 of our revision.
>
> > Bregman divergence over other divergence functions?
>
> We note that the Bregman Divergence is a generalization of many popular divergences, where choosing a suitable convex function $h$ in the Bregman Divergence recovers the desired divergence [4]. For example, if one chooses $h(x) = \frac{1}{2}||x||_2^2$ as we did in our manuscript, the Bregman Divergence becomes the (halved) squared Euclidean distance. With this choice of $h$, we are able to obtain a simple closed-form gradient that matches the steering updates. A full table of the different convex functions $h$ and the associated divergence can be found in [4], and modifying the choice of $h$ leads to new objectives and gradients, thus inducing new steering methods.
>
> > Intuition about how different choices of $h$ in the Bregman divergence lead to different feature maps and steering behaviors
>
> When $h$ is chosen to be quadratic, we recover the standard difference-in-means steering vector, as shown in our manuscript. However, for non-quadratic choices of $h$, different behaviors can emerge depending on the structure of $h$. For instance, if $h$ is designed to grow more rapidly for large-magnitude coordinates, the resulting gradients will place greater weight on those dimensions. More generally, by selecting $h$ that induces nonlinear transformations of individual features, the method can reshape the activation space$-$effectively stretching, compressing, or reweighting the steering direction along particular axes$-$yielding steering vectors with properties that differ fundamentally from the simple mean-difference approach.
>
> **References**
>
> [1] Andy Arditi, Oscar Balcells Obeso, Aaquib Syed, Daniel Paleka, Nina Rimsky, Wes Gurnee, and Neel Nanda. Refusal in language models is mediated by a single direction. NeurIPS, 2024.
>
> [2] Hieu M. Vu and Tan Minh Nguyen. Angular steering: Behavior control via rotation in activation space. NeurIPS, 2025.
>
> [3] Pau Rodriguez, Arno Blaas, Michal Klein, Luca Zappella, Nicholas Apostoloff, marco cuturi, and Xavier Suau. Controlling language and diffusion models by transporting activations. ICLR, 2025.
>
> [4] Arindam Banerjee, Srujana Merugu, Inderjit S. Dhillon, and Joydeep Ghosh. Clustering with Bregman Divergences. JMLR, 2005.

---

### Author Response · Authors · 2025-12-01
**To New AC - Empirical and Theoretical Findings Added During the Rebuttal Phase (1)**

Dear new AC,

In this message, we would like to summarize the newly obtained empirical and theoretical results obtained during this rebuttal period. The changes are also added to the revised manuscript and highlighted in Magenta.

**1. Additional Ablation Study on the Momentum Coefficient ($\beta$):**

To supplement the jailbreaking experiment in Section 4.1 and Appendix C, we performed an additional ablation study on the momentum coeefficient $\beta$ used in Momentum Steering, on Gemma2-9B-Instruct and Gemma2-27B-Instruct repsecitvely. The settings are identical to those described in Section 4.1, but instead of $\beta=0.99$, we vary $\beta$ between $0$ and $0.99$. Here, $\beta=0$ indicates that no Momentum Steering is used, and the new empirical results can found in Table 5 and 8 in Section 4.4 and Appendix D respectively in the revised manuscript. For convenience, we also present the results in a combined table in Table 1:

Table 1: Ablation study on different choices of momentum coefficient $\beta$. We report the ASR for each choice of $\beta$. Setting $\beta=0$ indicates no momentum and the experiments in Section 4.1 and Appendix C utilize $\beta=0.99$.

|   Method  &nbsp; &nbsp; &nbsp; &nbsp;| &nbsp; $\beta=0$ &nbsp; | &nbsp; $\beta=0.5$ &nbsp; | &nbsp; $\beta=0.75$ &nbsp; | &nbsp; $\beta=0.9$ &nbsp; | &nbsp; $\beta=0.95$ &nbsp; | &nbsp; $\beta=0.97$ &nbsp; | &nbsp; $\beta=0.99$ &nbsp; |
|:----------------|:-------------:|:-------------:|:-------------:|:-------------:|:-------------:|:-------------:|:-------------:|
|Gemma2-9B-Instruct|
| AS + Mom. | $7.69$ |$9.62$ | $26.92$ | $40.38$ | $46.15$| $45.19$ |$40.38$|
| AS + Mom. (AA) | $20.19$ |$21.15$ | $42.31$ | $47.12$ | $50.00$| $43.27$ |$42.31$|
| AS + Mom. (DA) | $19.23$ |$17.31$ | $32.69$ | $44.23$ | $50.96$| $44.23$ |$41.34$|
|Gemma2-27B-Instruct|
| AS + Mom. | $4.81$ |$5.77$ | $4.81$ | $6.73$ | $17.31$| $34.62$ |$71.15$|
| AS + Mom. (AA) | $6.73$ |$8.65$ | $7.69$ | $35.58$ | $76.92$| $59.62$ |$52.88$|
| AS + Mom. (DA) | $34.62$ |$42.31$ | $19.23$ | $44.23$ | $48.08$| $42.31$ |$45.19$|

---

> ### Author Response · Authors · 2025-12-01
> **To New AC - Empirical and Theoretical Findings Added During the Rebuttal Phase (2)**
>
> **2. Revision to our initial Stability Analysis in Appendix D in the original manuscript:**
>
> Following the suggestion of Reviewer vAxe, we have revised the stability analysis, and can be located now in Appendix E of the revised manuscript. In our revised version, the constructed discrete dybamical system now includes a time-varying bias, and the derivation of the system better coincides with the  explicit choice of function $h$ in Section 3. Additionally, we provide a sufficient condition for the convergence of our method and its proof.
>
> **3. Compilation of Steered Responses with Momentum Steering:**
>
> We followed the suggestion of Reviewer vAxe and included qualitative examples of how momentum steering alters the model output in the revised manuscript. Specifically, we compiled sample generations from Gemma2-9B-Instruct, steered using Angular Steering [1] with and without Momentum Steering, respectively, on a prompt from the test set used in our jailbreaking experiment in Section 4.1. The responses are compiled in Table 9, Appendix F of our revised manuscript. To summarize the key observations, we note that when steering without Momentum Steering, we are unable to obtain an angle that can bypass the model's safety mechanisms. However, when steering with Momentum Seering, we can both obtain a successful jailbreaking response and observe changes in behavior at different steering angles, as observed in [1].
>
> **4. Additional Plots of Steering Vector Norms from Pretrained Models:**
>
> Following the feedback given by reviewer VGaF, in addition to the plot of the steering vector norms (when computed sequentially) from the randomly initialized model in Figure 2, Section 3, we extended our analysis to consider the steering vector norms of pretrained models as well. Namely, we plotted the steering vector norms of both Llama3.2-3B-Instruct and Llama3.1-8B-Instruct in Figure 4 and 5, Appendix G, respectively.
>
> **5. Additional Experiments with Activation Addition:**
>
> Based on the feedback provided by Reviewer xjsN, we have extended the jailbreaking experiment to consider ActAdd [2]. Specifically, we keep the experimental settings mainly the same as described in Section 4.1, but we use ActAdd [2] instead of Angular Steering [1] as the intervention during inference, and we perform the experiment with Qwen2.5-3B-Instruct and Llama3.2-3B-Instruct. We note that in ActAdd, we perform the intervention at only one extraction point during inference, and the exact location is a tunable hyperparameter. We report the results in Table 10, Appendix H in our revised manuscript, and we also compile the results here in Table 2 for convenience:
>
> Table 2: Performance of ActAdd with and without Momentum Steering. The Extraction Point indicates the location of intervention and the layer index starts from 0. The $\gamma$ indicates the strength of intervention. We note that the steering vectors in all configurations have been normalized to a unit vector prior to intervention during inference. $\beta$ indicates the momentum coefficient and $\beta = 0$ indicates regular difference-in-means.
>
> |   Method  &nbsp; &nbsp; | &nbsp; ASR $\uparrow$ &nbsp; | &nbsp; &nbsp; &nbsp; &nbsp; &nbsp; &nbsp; &nbsp; &nbsp; Extraction Point &nbsp;|&nbsp; &nbsp; $\gamma$ &nbsp; &nbsp;|&nbsp; &nbsp; $\beta$ &nbsp; &nbsp; |
> |:----------------|:-------------:|:-------------:|:-------------:|:-------------:|
> |Qwen2.5-3B-Instruct|
> | ActAdd (Baseline) |     $49.04$    |       Layer 19, Input LayerNorm| $60$ | $0$ |
> | ActAdd + Mom. | $65.38$| Layer 18, Post Attention LayerNorm | $42.5$ | $0.99$ |
> |Llama3.2-3B-Instruct|
> | ActAdd (Baseline) |    $60.58$     |        Layer 16, Input LayerNorm | $20$ | $0$|
> | ActAdd + Mom. | $62.50$ | Layer 16, Input LayerNorm | $22.5$ | $0.5$ |

---

> ### Author Response · Authors · 2025-12-01
> **To New AC - Empirical and Theoretical Findings Added During the Rebuttal Phase (3)**
>
> **6. Improvement of Adam Steering on Gemma2-27B-Instruct:**
>
> In Table 6 (Table 7 in the revised manuscript), Appendix C, of our original manuscript, under the original choice of moment coefficients ($\beta_1=0.9$, $\beta_2=0.999$), we notice the suboptimal performance of Adam Steering on the jailbreaking task. However, upon changing to a new set of moment coefficients ($\beta_1=0.999$, $\beta_2=0.5$), we were able to observe significant performance gains as compared to baseline. The new results have been updated in Table 7, Appendix C, highlighted in Magenta in the revised manuscript.
>
> **7. Additional Experiments with Top-K Sampling:**
>
> Following the feedback from reviewer VGaF, we have also extended the jailbreaking experiment in Section 4.1 to consider Top-K Sampling (with $K=10,40$ respectively) during inference instead of just greedy-decoding. We performed the experiment with Gemma2-9B-Instruct and Gemma2-27B-Instruct respectively, and for each configuration of model and $K$, we ran the experiments 10 times. The results for this experiment can be found in Table 11, Appendix I in the revised manuscript, and we also compile the results here in Table 3 for convenience:
>
> Table 3: Performance of all configurations of our method with Gemma2-9B-Instruct and Gemma2-27B-Instruct on the jailbreaking task in Section 4.1, with Top-K ($K = 10, 40$ respectively) sampling over 10 runs. We report the mean and standard deviation of the ASR across the 10 runs for each configuration.
>
> |   Method  &nbsp; &nbsp;|&nbsp; &nbsp; &nbsp; Seq. &nbsp; &nbsp; &nbsp;|&nbsp; &nbsp; ASR $\uparrow$ &nbsp;|
> |:----------------|:-------------:|:-------------:|
> |Gemma2-9B-Instruct, $K = 10$|
> | AS (Baseline) |         |       $6.73\scriptstyle{\pm 1.11}$ |
> | AS + Mom. | | $39.71\scriptstyle{\pm 4.04}$ |
> | AS + Mom. (AA) | $\checkmark$| $40.29\scriptstyle{\pm 4.61}$ |
> | AS + Mom. (DA) | $\checkmark$| $37.69\scriptstyle{\pm 3.59}$ |
> | AS + Adam | | $29.52\scriptstyle{\pm 2.65}$|
> | AS + Adam (AA) | $\checkmark$| $34.04\scriptstyle{\pm 3.46}$ |
> | AS + Adam (DA) | $\checkmark$| $24.90\scriptstyle{\pm 2.65}$ |
> |Gemma2-9B-Instruct, $K = 40$|
> | AS (Baseline) |         |        $6.44\scriptstyle{\pm 0.65}$ |
> | AS + Mom. | | $38.94\scriptstyle{\pm 5.01}$ |
> | AS + Mom. (AA) | $\checkmark$| $40.96\scriptstyle{\pm 4.18}$ |
> | AS + Mom. (DA) | $\checkmark$| $37.79\scriptstyle{\pm 3.17}$ |
> | AS + Adam | | $29.13\scriptstyle{\pm 3.21}$ |
> | AS + Adam (AA) | $\checkmark$| $34.33\scriptstyle{\pm 4.23}$ |
> | AS + Adam (DA) | $\checkmark$| $25.29\scriptstyle{\pm 5.03}$ |
> |Gemma2-27B-Instruct, $K = 10$|
> | AS (Baseline) |         |       $4.42\scriptstyle{\pm 1.22}$ |
> | AS + Mom. | | $67.40\scriptstyle{\pm 2.74}$ |
> | AS + Mom. (AA) | $\checkmark$| $49.81\scriptstyle{\pm 4.22}$ |
> | AS + Mom. (DA) | $\checkmark$| $43.46\scriptstyle{\pm 3.39}$ |
> | AS + Adam | | $49.81\scriptstyle{\pm 3.29}$|
> | AS + Adam (AA) | $\checkmark$| $26.83\scriptstyle{\pm 3.49}$ |
> | AS + Adam (DA) | $\checkmark$| $31.35\scriptstyle{\pm 3.27}$ |
> |Gemma2-27B-Instruct, $K = 40$|
> | AS (Baseline) |         |        $4.81\scriptstyle{\pm 1.28}$ |
> | AS + Mom. | | $67.50\scriptstyle{\pm3.32}$ |
> | AS + Mom. (AA) | $\checkmark$| $49.52\scriptstyle{\pm 3.02}$ |
> | AS + Mom. (DA) | $\checkmark$| $44.13\scriptstyle{\pm 3.25}$ |
> | AS + Adam | | $51.63\scriptstyle{\pm 2.72}$ |
> | AS + Adam (AA) | $\checkmark$| $26.63\scriptstyle{\pm 3.51}$ |
> | AS + Adam (DA) | $\checkmark$| $31.63\scriptstyle{\pm 2.29}$ |

---

> > ### Author Response · Authors · 2025-12-01
> > **To New AC - Empirical and Theoretical Findings Added During the Rebuttal Phase (4)**
> >
> > In addition to the above, we would also like to mention, at the suggestion of our reviewers, the other changes made to strengthen our paper:
> > 1. Updated Section 2 and 3 to better improve the presentation and clarity of our notation, background, and theory related to our optimization framework.
> > 2. Updated the section containing the missing citations to include the relevant citations, as pointed out by reviewer xjsN.
> > 3. Fixed the typos pointed out by the reviewers and made minor revisions across the paper to better improve the clarity.
> >
> >
> > **References**
> >
> > [1] Hieu M. Vu and Tan Minh Nguyen. Angular steering: Behavior control via rotation in activation space. NeurIPS, 2025.
> >
> > [2] Alexander Matt Turner, Lisa Thiergart, Gavin Leech, David Udell, Juan J Vazquez, Ulisse Mini, Monte MacDiarmid. Steering Language Models with Activation Engineering. OpenReview preprint, 2023.

---

### Author Response · Authors · 2025-12-01
**To New AC: We Appreciate Your Willingness to Oversee Our Submission**

Dear new AC,

We are grateful that you have stepped in at this point in the review cycle to assume responsibility for our submission. Your time and attention are sincerely appreciated.

To facilitate your evaluation of both the paper and its review history, we have organized three concise messages, which appear in the comments that follow:

*Message 1. Summary of Novelties and Contributions of Our Work.*

*Message 2. Summary of Primary Reviewer Questions and Our Responses.*

*Message 3. Empirical and Theoretical Findings Added During the Rebuttal Phase.*

We submitted our initial rebuttal on November 18 (AoE) and followed up with additional experimental results on November 22 (AoE). Despite several reminders, the reviewers did not provide any further responses, and the discussion period concluded earlier than anticipated. Consequently, we were unable to obtain additional clarification or updated feedback from them.

We kindly ask that you review our paper, rebuttal, and subsequent responses (including the new results) and consider that we have thoroughly addressed all of the reviewers’ concerns. We believe that, had the discussion continued, the reviewers would likely have revised their assessments accordingly.

Thank you once again for overseeing our submission. If any aspect of the paper or rebuttal requires clarification, please do not hesitate to contact us. We would be glad to provide further details or engage in additional discussion.

Warm regards,

Authors

---

### Meta-Review · Area_Chair_mj84 · 2026-01-08

**Summary:**

While the paper proposes an original framework reinterpreting activation steering as an optimization problem, reviewers raised consistent concerns that inform a rejection decision. First, the work does not meet expected standards for clarity and technical rigor. Multiple reviewers highlighted that the manuscript suffers from presentation issues, including inconsistent notations, typos, and unexplained mathematical terms or designs. Key theoretical steps, e.g., specifically the reduction of steering to projected gradient descent, remain difficult to follow. Furthermore, the proposed framework lacks sufficient justification for its design choices, such as the use of Bregman divergence or the hypothesis that layer functions act as projections. Consequently, the current state of the manuscript falls below publication standards.

**Reviewer Concerns:**

The rebuttal clarified the "offline" nature of the momentum precomputation, addressing concerns regarding inference-time computational overhead. However, it did not meaningfully resolve the core technical and clarity issues identified in the reviews. Concerns about inconsistent notation and the lack of justification for technical choices stand. While the authors added experimental results and refine part of presentations during the rebuttal, I believe the paper needs more substantial revision to improve the overall clarity and bridge the gap in theoretical formalization.

**Reviewer Scores:**

Given the presentation issues at its current state, no positive score changes are expected.

---

### Decision · Program_Chairs · 2026-01-26

Reject